# Monitoring *Leishmania* *infantum* Infections in Female *Lutzomyia* *longipalpis* by Using DNA Extraction on Cation Exchange Paper and PCR Pool Testing

**DOI:** 10.3390/diagnostics12112653

**Published:** 2022-11-01

**Authors:** Tiago Leonetti Coutinho, Fernando Augusto Lima Marson, Osias Rangel, Selma Giorgio, Kamila Cristina Silva, Carlos Emilio Levy

**Affiliations:** 1Microbiology Laboratory, Department of Clinical Pathology, Faculty of Medical Sciences, University of Campinas, Campinas 13083-887, São Paulo, Brazil; 2Post Graduate Program in Health Science, Laboratory of Medical and Human Genetics, São Francisco University, Bragança Paulista 12916-900, São Paulo, Brazil; 3Pasteur Institute, São Paulo 01311-000, São Paulo, Brazil; 4Biology Institute, University of Campinas, Campinas 13083-970, São Paulo, Brazil

**Keywords:** diagnosis, epidemiology, *Leishmania infantum*, *Lutzomyia longipalpis*, minimum infection rate, polymerase chain reaction, visceral leishmaniasis

## Abstract

Visceral leishmaniasis remains a serious public health issue, and Brazil was among the seven countries with the highest prevalence of this disease worldwide. The measures to control this disease are not easily developed, and the improvement of its diagnosis, surveillance, and control is still needed. This study aimed to carry out the polymerase chain reaction (PCR) diagnosis of *Leishmania infantum* in vector samples in some municipalities of the State of São Paulo, which included two municipalities with human disease transmission and two with dog transmission only. Vectors were collected in traps with luminous bait. Next, they were killed at −4 °C and kept in 70% alcohol. Groups of ten female insects (pools) were mashed on cation exchange paper (fine cellulose phosphate with 18 µEq/cm² ionic exchange capacity) for DNA extraction. The PCR was carried out to identify the natural infection of the *Leishmania* genus in female *Lutzomyia longipalpis* (*Lu. Longipalpis*). Out of the 3,880 *Lu. longipalpis* phlebotomines, 1060 were female and 2820 were male (3:1). The method used to extract the DNA in pools of ten phlebotomines and the PCR resulted in sensitivity, specificity, practicality, and faster analyses when compared to the individual analysis method. The procedure described can be used on a large scale in the leishmaniasis epidemiological surveillance, enabling a higher number of analyses and the optimization of human resources because the traditional diagnostic method is carried out via desiccation of the insect digestive system and microscopic examination, which is time-demanding and there is the need of manual skills.

## 1. Introduction

Leishmaniases are neglected zoonoses with a wide clinical spectrum and a great variety of parasites, reservoirs, and vectors involved in their transmission [1]. Over 90% of the visceral leishmaniasis cases in the world were reported in seven countries, namely, Brazil, Ethiopia, India, Kenia, Somalia, South Sudan, and Sudan [1].

Visceral leishmaniasis has been a public health issue in the municipalities of the State of São Paulo since the late 1990s [2]. That state, in the period from 1999 to 2018, totaled 7931 reported cases of visceral leishmaniasis in humans, out of which 2953 (37.2%) were confirmed as native and distributed in 103 (15.9%) municipalities. The number of deaths reported was 255, resulting in an 8.6% lethality rate in this region [2].

Visceral leishmaniasis found in Europe, Africa, and the Americas is caused by *Leishmania infantum* (*L. infantum*) [1]. In Brazil, the main vector is *Lutzomyia*
*longipalpis* (*Lu. longipalpis*), and the domestic dog is its main reservoir [3]. Regarding the vector, its presence was identified in 202 municipalities in the State of São Paulo, and 151 (74.8%) transmissions were also observed. *Lu. longipalpis* was found in all 97 municipalities with both canine and human transmission [2].

The diagnosis of parasitological natural infection by *Lu. longipalpis* is a vital component in the leishmaniosis surveillance and control program. It is carried out via desiccation of the insect digestive system and microscopic examination, whose mean positivity rates range between 0 and 0.9%. However, this technique is hard to operationalize, and new strategies are required that might include molecular methods based on polymerase chain reaction (PCR), resulting in higher sensitivity and specificity [4,5,6,7,8,9]. In such a context, the objective of this study was to carry out *L. infantum* diagnosis in *Lu. longipalpis* samples using PCR in the State of São Paulo, Brazil, and standardize a practical and faster method of extraction of the genomic material using a cation exchange paper that can be used on a large scale.

## 2. Materials and Methods

Entomological collections of *Lu. longipalpis* samples were carried out in two municipalities (Araçatuba, located in the western portion of the State of São Paulo, latitude 21°12′32″ S, longitude 50°25′58″ W; and Dracena, also located in the western São Paulo State, latitude 21°28′51 S, longitude 51°51′58″ W), classified by the State Health Secretariat of São Paulo as visceral leishmaniasis perennial transmission cities. Two municipalities of the administrative region of Campinas (Valinhos, latitude 22°58′16″ S, longitude 46°59′47″ W; and São Pedro, latitude 22°32′55″ S, longitude 47°54′50″ W) were also investigated and classified as dog reservoir sites only, where the transmission to humans was not yet evident (Figure 1). In our study cities, the *Lu. longipalpis* is the main vector of *Leishmania*. In addition, among the collected phlebotomines, the selection/identification of the specie *Lu. longipalpis* was conducted by two authors (T.L.C. and O.R.) with expertise in entomology.

The capture was carried out using four traps for two years (2017 and 2018) during the hottest months of the year (January to May and September to December) in households that presented favorable features to the presence of phlebotomines (*Lu. longipalpis*). Luminous bait traps of the CDC, Sudia, and Chamberlaim 1962 types were exposed for 12 h in the households.

The phlebotomine samples (*Lu. longipalpis*) were transported in the trap collectors to the laboratory. Next, the insects were killed at −4 °C and kept in 70% alcohol. After the triage of the captured samples, the phlebotomines (*Lu. longipalpis*) were selected and separated according to their sex. A pool of ten females per tube was considered as a sample for genetic analysis. The pools of sand flies were handled individually and the maceration of the sand flies, as well as the DNA extraction, was performed one at a time (pools) to avoid contamination between samples. All the permanent materials used during the procedures were sterilized after the protocol before being reused. In addition, disposable materials such as tubes, tips, and gloves were discarded. All the procedures were accompanied by professionals with extensive experience in the clinical analysis laboratory of a reference center—University Hospital.

Each sample was crushed using tips on cation exchange paper (fine cellulose phosphate with 2.5 cm diameter and 0.23 mm thickness, 18 µEq/cm² ionic exchange capacity, and 125 mm flow at each 30 min) measuring 0.5 × 3 cm, brand Whatman cation exchanger (46 × 57 cm). The cation exchange paper containing the phlebotomines was kept at 37 °C for 15 min, and the excess carcass was removed from the paper. Next, the paper containing the material was homogenized in a vortex in 500 µL of ultrapure water for 20 s and centrifuged at 10,000 r.p.m. for 15 min. After centrifugation, the remaining paper was removed, and the material was subjected to DNA extraction.

The DNA extraction was carried out using the DNeasy Blood and Tissue commercial kit (Qiagen, Hilden, Germany), following the manufacturer’s instructions.

The PCR reactions were obtained using the GoTaq^®^ Green Master Mix commercial kit (Promega, Madison, WI, USA) containing the following reagents: MgCl_2_, enzymatic buffer, deoxynucleotide triphosphates (dNTP), and Taq DNA polymerase. The reaction final volume was 20 µL, containing: seven µL Milli-Q sterile water, 12.5 µL GoTaq^®^ Green Master Mix, and 25 pmol of each primer. 

The *Leishmania* spp. identification was carried out by the Leish-150 [5′-GGG(G/T)AGGGGCGTTCT(C/G)CGAA-3′] and Leish-152 [5′-(C/G)(C/G)(C/G)(A/T)CTAT(A/T)TTACACCAACCCC-3′] primers at the 55 °C annealing temperature and amplification of a DNA fragment of 120 base pairs (bp) [12]. 

To identify the *L. infantum* specie, the RV1 (5’-CTTTTCTGGTCCCGCGGGTAGG-3´) and RV2 (5´—CCACCTGGCCTATTTTACACCA-3´) primers were used at the 55 °C annealing temperature and amplification of a fragment with 145 bp. 

To investigate the presence of phlebotomine DNA (*Lu**. longipalpis*), an amplification of the DNA fragment with 370 bp in the 28S ribosomal region was carried out using the Lu1 (5´-TGAGCTTGACTCTAGTTTGGCAC-3´) and Lu2 (5´-AGATGTACCGCCCCAGTCAAA-3´) primers. These primers were used in the reactions as endogenous control.

Negative controls were added in the analyses (nuclease-free water and GoTaq^®^ Green Master Mix commercial kit), positive control for the *L. infantum* specie (DNA extracted from promastigote form, strain reference: MHOM/BR/73/M2269), and Lu1 and Lu2 endogenous control to confirm the DNA extraction (male insects captured in the field). The inclusion of uninfected male insects allowed the contamination control since these samples cannot contain the *Leishmania* spp. DNA. 

The samples were amplified in a thermocycler (Eppendorf, São Paulo, Brazil) with initial denaturation at 95 °C for five minutes, followed by 29 cycles at 95 °C for 45 s for denaturation, 55 °C for 45 s for annealing, 72 °C for 45 s for the DNA fragment extension, and for the final extension, a cycle at 72 °C for five minutes was carried out. Electrophoresis was carried out in agarose gel (2%) in a 0.5X TBE buffer (10X TBE buffer tris-borate-EDTA, pH 8.0) stained with ethidium bromide. The PCR was subjected to electrophoresis for 30 min at 100 V and 40 mA in 0.5X TBE buffer and then, photographed in a transilluminator (Loccus-biotecnologia, Cotia, São Paulo, Brazil).

## 3. Results

In the study, 3880 phlebotomines (*Lu. longipalpis*) were collected, out of which 1060 were female and 2820 were male. The animal distribution according to the municipalities investigated and sex is shown above (Table 1). The predominance of male insects over females was observed in a 3:1 proportion. 

Initially, we used two types of paper filters, an anion exchanger and a cation exchanger, which were immersed with different volumes of the *Leishmania* culture in the initial concentration of 5 × 10^7^ parasites per mL. On the anion exchange paper, the band intensity had a lower intensity than that of the cation exchange paper (Figure 2A,B). The latter presented positive PCR to *L. infantum* in all the concentrations used (Figure 2B). In this experiment, to identify the PCR technical detection limit, the cation exchange paper was contaminated with small amounts of *Leishmania* from the initial concentration of 5 × 10^7^ parasites per mL (0.125 µL–6.25 × 10^3^ parasites, 0.25 µL–1.25 × 10^4^ parasites, 0.5 µL–2.5 × 10^4^ parasites, 0.75 µL–3.75 × 10^4^ parasites, 1 µL–5 × 10^4^ parasites, 5 µL–2.5 × 10^5^ parasites, and 10 µL–10 × 10^5^ parasites) (Figure 2), and its sensitivity to detect concentrations ≥10^3^ was demonstrated, which is above the number of parasites (*L. infantum*) per mL in natural sand flies (phlebotomines, *Lu. longipalpis*) after the infection where it was identified +10^5^ parasites per mL [13].

The 1060 female insects captured were grouped in pools of ten insects each and according to the city. A hundred and six pools were analyzed; and two pools presented positive results, one from Araçatuba and one from Dracena (Table 1). The positive samples had their PCR reactions repeated to confirm the results. A PCR was carried out to detect the species, which was described as *L. infantum* (Figure 2C).

The minimum infection rate was calculated considering all the samples, with ten individuals (pool) and a single phlebotomine (*Lu. longipalpis*) positive for *Leishmania* per the sample (Table 1). The formula used to calculate the minimum infection rate was the number of positive groups (pools) × 100/total phlebotomines processed [10,11]. The minimum infection rate in all the cities was simultaneously 0.19%. The minimum infection rate in Araçatuba and Dracena was 0.22%. 

*L. infantum* was tested at the specie level because it is responsible for causing visceral leishmaniasis in Brazil. We also tested the *Leishmania* genus to demonstrate that the technique can be applied in other places where different species are liable to cause the disease.

## 4. Discussion

The determination of natural infection in vector species, mainly in endemic regions, and the correct identification of the *Leishmania* species in phlebotomines (*Lu. longipalpis*) are highly relevant in the epidemiology of leishmaniases [10,14], mainly in Brazil, where the intense infection is yet described.

In this study, 100% of the female insects collected were screened and subjected to a diagnosis of natural infection using the PCR, which analyzed 106 female phlebotomine pools (*Lu. longipalpis*). The results obtained revealed two positive pools from the total of samples processed with a 0.22% infection minimum rate. Considering the human transmission areas, the percentage of positivity of the pools observed in this study (2/92 = 2.2%) agreed with the report put forward by Michalsky et al. (6/155 = 3.9%) [15]. Still, it was lower than that presented by Soares et al. (6/74 = 8.1%) [16]. 

Vexenat et al. reported that the investigation of infection in phlebotomines captured in a non-targeted manner showed a minimum infection rate even lower (or null) [17]. In endemic leishmaniasis areas, the natural minimum infection rate is usually considered low, not justifying the illness’s high transmission, mainly in areas recently occupied [18]. On the other hand, the investigation of natural *Leishmania* infection using the conventional method of female phlebotomine desiccation has produced infection rates that range between 0 and 9%. Still, these indices were below 1% in the different endemic areas investigated, in Brazil and other countries in South America [19]. 

It seems relevant to highlight the importance of PCR as a diagnostic tool in this study since its main advantages include greater sensitivity and specificity when compared to conventional methods (phlebotomine desiccation), and greater speed in the diagnosis of large samples. Among the species of vectors collected, taking into consideration their density in the same region in the State of São Paulo, *Lu. longipalpis* is largely known as a visceral leishmaniasis vector in Brazil. Thus, this study shows the *Lu. longipalpis* population capacity to become infected with *Leishmania*, indicating their probable role as a visceral leishmaniasis vector in the state. However, in some Brazilian municipalities, other species have taken part in visceral leishmaniasis transmission, such as in Pernambuco, where *Mygonemyia migonei* was recognized as a permissible species for the transmission of this disease [3]. In such cases, the development of PCR techniques plays a vital role in proving the natural infection as part of the attributes suggested by Killick-Kendrick to clarify the type of vector [20]. 

Our results revealed positive test results for municipalities with perennial human transmission and negative where only canine transmission was found. However, we had a low number of pools in the cities where only canine transmission was found; and only two pools were positive where perennial human transmission occurred. In such a context, our results demonstrated that DNA extraction and PCR were effective in evaluating the presence of the parasite; however, we cannot prove the perennial human transmission and canine transmission status in the cities where the sand fly specimens were collected due to several factors, such as the low number of samples collected.

The use of the PCR test might allow for fast evaluation of the risk of leishmaniasis occurrence, enabling the control of these vectors in endemic areas by epidemiological surveillance programs. The results obtained in this study showed that the infection circulates in phlebotomines in household environments; thus, with the molecular tools available, studies on natural *Leishmania* infection might be developed aiming to characterize transmission areas.

## 5. Conclusions

Phlebotomine infection by the *Leishmania* genus in the regions investigated in the State of São Paulo was seen to be active and perennial, with an intense transmission reported by the Brazilian health authorities. The method used to extract DNA from the pools of ten phlebotomines (cation exchange paper) and the PCR reactions were seen to be useful due to their sensitivity, specificity, practicality, and faster analysis when compared to the individual analysis method. This resource can be used on a large scale in the epidemiological surveillance of leishmaniasis, enabling a higher number of analyses and the optimization of human resources.

## Figures and Tables

**Figure 1 diagnostics-12-02653-f001:**
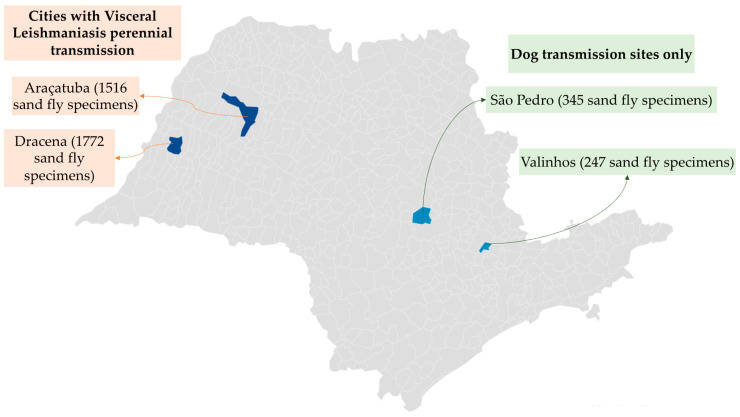
Distribution of *Lutzomyia longipalpis* (animals) samples collected in the State of São Paulo in Brazil according to the cities with visceral leishmaniasis perennial transmission and the presence of dog reservoir sites only. The samples were used to carry out the DNA extraction and the polymerase chain reaction (PCR) analysis by pools of ten animals. The sex of the animals is presented in Table 1.

**Figure 2 diagnostics-12-02653-f002:**
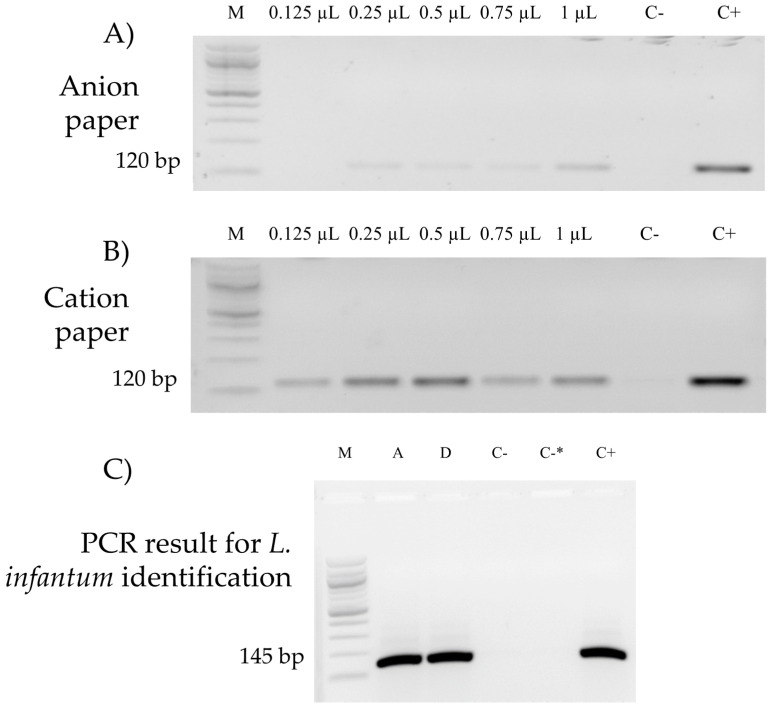
(**A**) Polymerase chain reaction (PCR) results to evaluate the DNA extraction quality using anion exchange paper. (**B**) PCR results to evaluate the DNA extraction quality using cation exchange paper. (**A**,**B**): M, 100 basis pair (bp) molecular weight marker (100 bp DNA Ladder), and the DNA fragments obtained after the DNA amplification for *Leishmania* spp. identification (120 bp); C-, PCR reaction negative control; C+, PCR reaction positive control with *Leishmania infantum* DNA extracted from cell culture. (**C**) PCR reaction results to evaluate natural infection of phlebotomines by *L. infantum*. M, 100 bp molecular weight marker (100 bp DNA Ladder) and the DNA fragments obtained after the DNA amplification for *L. infantum* specie (145 bp) identification; A, female phlebotomines subjected to molecular diagnosis, revealing positivity in Araçatuba; D, positive females in Dracena; C-, PCR reaction negative control; C-*, PCR reaction negative control containing DNA extracted from male phlebotomines; C+, PCR reaction positive control (DNA of a pool of ten male phlebotomines mixed with *L. infantum* DNA extracted from cell culture).

**Table 1 diagnostics-12-02653-t001:** Distribution of individuals collected in absolute and percentage numbers of *Lutzomyia longipalpis* per sex and municipalities investigated in the State of São Paulo, Brazil. The absolute number of pools was analyzed with positive results, and an infection minimum rate was presented.

Municipality	Female *		Male *		Total		Pools (N = 10)		Infection Minimum Rate (%) **
	N	%	N	%	N	%	N	Positive	
Araçatuba	460	30.3	1056	69.7	1516	100	46	1	0.22
Dracena	460	25.9	1312	74.1	1772	100	46	1	0.22
São Pedro	80	23.2	265	76.8	345	100	8	0	0.00
Valinhos	60	24.3	187	75.7	247	100	6	0	0.00
Total	1060	27.3	2820	72.7	3880	100	106	2	0.19

N, number of individuals; % percentage. *, the percentage of females and males was calculated by the following formula = (number of females or males)/(total number of *Lutzomyia longipalpis* collected in each municipality). **, the infection minimum rate was calculated using the following formula: the number of positive groups (pools) × 100/Total phlebotomines processed [10,11].

## Data Availability

Not applicable.

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
