# Peer review of "Monitoring Leishmania infantum Infections in Female Lutzomyia longipalpis by Using DNA Extraction on Cation Exchange Paper and PCR Pool Testing"

_diagnostics, 2022, doi:10.3390/diagnostics12112653_

Round 1

Reviewer 1 Report

The study was aimed to develop a monitoring of Leishmania infection in sand flies. This kind of study was well-established, and unfortunately, I could not find any improvement in this study. In addition, several serious issues were found in the methodology.

1.     The DNA extraction process is complicated. I’m afraid it has more risk for contamination.

2.     The sensitivity is very low (> 103, line 132), and not suitable for monitoring Leishmania infection in sad flies.

3.     How did you identify sand flies? The authors described sand flies captured were all Lutzomyia longipalpis: however, it is not possible because several sand fly species are distributing in endemic areas.

4.     What is “mosquitoes” (lines 26 and 87)?

5.     I totally disagree “the protocol is sensitive and lower cost, and faster” (line 35). Again, I cannot find any advantage to use this protocol for the monitoring.

Author Response

Comments and Suggestions for Authors 1

The study was aimed to develop a monitoring of Leishmania infection in sand flies. This kind of study was well-established, and unfortunately, I could not find any improvement in this study. In addition, several serious issues were found in the methodology.

Reply: Dear reviewer, we thank you for your significant contribution to revising our paper. We replied to all comments aiming to solve the major questions raised by you. Also, we had several methods to screen the Leishmania published in the literature, mainly associated with parasite identification through dissect action. However, in Brazil, we had a low number of technicians trained to perform such techniques, which is time-demanding. Also, we had the improvement of availability of PCR to screen the parasite. In this context, we included a new method to isolate the DNA from the study sample, and we performed the genetic screening using primers previously published.

  1. The DNA extraction process is complicated. I’m afraid it has more risk for contamination.

            Dear reviewer, we thank you for your essential contribution. In daily practice, the protocol is fast and straightforward to handle. Regarding the risk of contamination, we included the following statement in the text:

The pools of sand flies were handled individually, and the maceration of the sand flies, as well as the DNA extraction, was performed one at a time (pools) to avoid contamination between samples. All the permanent materials used during the procedures were sterilized after the protocol before being reused. In addition, disposable materials such as tubes, tips, and gloves were discarded. All the procedures were accompanied by professionals with extensive experience in the clinical analysis laboratory of a reference center – University Hospital.

  1. The sensitivity is very low (>103, line 132), and not suitable for monitoring Leishmania infection in sad flies.

Reply: The authors thank the reviewer, and we included the following statement:

In this experiment, to identify the PCR technical detection limit, the Cation paper was contaminated with small amounts of Leishmania from the initial concentration of 5 x 107 parasites per mL (0.125 µL – 6.25 x 103 parasites, 0.25 µL – 1.25 x 104 parasites, 0.5 µL – 2.5 x 104 parasites, 0.75 µL – 3.75 x 104 parasites, 1 µL – 5 x 104 parasites, 5 µL – 2.5 x 105 parasites, and 10 µL – 10 x 105 parasites) (Figure 2), and its sensitivity to detect concentrations ≥103 was demonstrated which is above the number of parasites (L. infantum) per mL in natural sand flies (phlebotomines, Lu. longipalpis) after the infection where it was identified +105 parasites per mL [10].

  1. How did you identify sand flies? The authors described sand flies captured were all Lutzomyia longipalpis: however, it is not possible because several sand fly species are distributing in endemic areas.

            Reply: The authors thank the reviewer, and we included the following statement in the methods:

In our study cities, the Lu. longipalpis is the main vector of Leishmania. Also, among the collected phlebotomines, the selection/identification of the specie Lu. longipalpis was done by two authors (T.L.C. and O.R.) with expertise in entomology.

  1. What is “mosquitoes” (lines 26 and 87)?

Reply: We thank the reviewer. We corrected the term to “phlebotomines”.

  1. I totally disagree “the protocol is sensitive and lower cost, and faster” (line 35). Again, I cannot find any advantage to use this protocol for the monitoring.

Reply: Dear reviewer, we performed some corrections in the text as we decided to include only a minor modification in the excerpt described by you as follows:

The method used to extract the DNA in pools of ten phlebotomines and the PCR resulted in sensitivity, specificity, practicality, and faster analyses when compared to the individual analysis method.

The PCR showed great sensitivity and specificity due to the remarkable capacity to identify a low amount of DNA from the Leishmaniaachieved after the extraction from the L. longipalpis sand flies. Also, the study protocol is easily handled when compared with the individual analysis method (desiccation of the insect digestive system and microscopic examination), which is time-demanding.

In addition, we removed the citation of cost because both protocols (DNA extraction/PCR and individual analysis) had different prices, and it was not evaluated in the present study. 

Reviewer 2 Report

While this is a very interesting approach to diagnose potential Leishmania spread in populations, I am unclear about several aspects of this research as indicated below:

Line 24: for the word ‘died’, do the authors mean ‘dried’ or ‘killed’ ?

In title and line 25 (as well as other places in the manuscript) the authors use the term cation paper and I am unclear what this means; the authors indicate cation paper (line 25) is fine cellulose phosphate and do not indicate a pH.  Since phosphate has a negative charge, it then should bind cations (which have positive charge) , but this will be pH dependent since at acidic pH the phosphate group becomes protonated and loses its charge.  Since nucleic acids also have negative charge (again as a function of pH), it is unclear to me why DNA will bind to the cellulose phosphate paper.  The authors need to give more information about the method used especially in relation to the solvent (buffer??) used and the pH. 

The inclusion of uninfected male insects was a good control to use.

Line 121:  to what does ‘Ludwig’ refer? 

Line 126:  is the word ‘contaminated’ really the correct word choice?

Line 132:  what is the numeral and units for concentration?

For the data in Figure A,B, and C:  please indicate the size (along the y axis) of the markers (using bp); in the legend, it indicates that 100 bp marker was used…does this indicate that the visible spots in the electrophoresis are 10-20 bp?  Please indicate that Figure 2 are results of electrophoresis.  Indicate the size of the spots to the right of the gel.

I am very unclear how the percentages were calculated (line 146 and in Table 1).  This should be clarified. 

Line 208: add ‘thus’ following the word environments to make this sentence clearer.

Line 213:  the data are not sorted by year so not sure how the authors can indicate ‘perennial’.

Line214:  replace ‘was’ with ‘were’

Line 215: the authors give no information about cost in this manuscript so not sure they can indicate ‘lower cost’.  A cost analysis is thus needed to make this statement.

In addition, for the title, change ‘in’ to ‘using’

The genus Leishmania should be in italics every where in the manuscript

Should ‘Leishmaniasis’ be capitalized or lower case throughout the manuscript?  ….should be consistent.

Author Response

Comments and Suggestions for Authors 2

While this is a very interesting approach to diagnose potential Leishmania spread in populations, I am unclear about several aspects of this research as indicated below:

Line 24: for the word ‘died’, do the authors mean ‘dried’ or ‘killed’?

Reply: We corrected the term to “Killed”.

In title and line 25 (as well as other places in the manuscript) the authors use the term cation paper and I am unclear what this means; the authors indicate cation paper (line 25) is fine cellulose phosphate and do not indicate a pH.  Since phosphate has a negative charge, it then should bind cations (which have positive charge), but this will be pH dependent since at acidic pH the phosphate group becomes protonated and loses its charge.  Since nucleic acids also have a negative charge (again as a function of pH), it is unclear to me why DNA will bind to the cellulose phosphate paper.  The authors need to give more information about the method used especially in relation to the solvent (buffer??) used and the pH.  

Reply: Dear Reviewer, we corrected the text to improve the understanding. We used the Cation paper (negative charge) to repel the DNA from the cells. Also, the pH was neutral. The following information was added:

Each sample was crushed using tips on Cation paper (fine cellulose phosphate with 2.5 cm diameter and 0.23 mm thickness, 18 µEq/cm² ionic exchange capacity, and 125 mm flow at each 30 min) measuring 0.5 x 3 cm, brand Whatman Cation Exchanger (46 x 57 cm). The Cation paper is negatively charged, which promotes the repelling of the Leishmania spp. DNA from the cells. After each phase, the Cation paper containing the phlebotomines was kept at 37 ºC for 15 min, and the excess carcass was removed from the paper. Next, the paper containing the material was homogenized in a vortex in 500 µL of ultrapure water for 20 sec and centrifuged at 10,000 r.p.m. for 15 min. After centrifugation, the remaining paper was removed, and the material was subjected to DNA extraction.

The DNA extraction was carried out using the DNeasy Blood and Tissue commercial kit (Qiagen, Hilden, Germany), following the manufacturer’s instructions.

The PCR reactions were obtained using the GoTaq® Green Master Mix commercial kit (Promega, Madison, Wisconsin, USA) containing the following reagents: MgCl2, enzymatic buffer, deoxynucleotide triphosphates (dNTP), and Taq DNA polymerase. The reaction final volume was 20 µL, containing: seven µL Milli-Q sterile water, 12.5 µL GoTaq® Green Master Mix, and 25 pmol of each primer.

The inclusion of uninfected male insects was a good control to use.

Reply: The authors thank the reviewer for the important feedback.

Line 121:  to what does ‘Ludwig’ refer?  

Reply: We deleted the term.

Line 126:  is the word ‘contaminated’ really the correct word choice?

Reply: We changed the term to “immersed”.

Line 132:  what is the numeral and units for concentration?

Reply: The authors thank the reviewer, and we included the following statement:

In this experiment, to identify the PCR technical detection limit, the Cation paper was contaminated with small amounts of Leishmania from the initial concentration of 5 x 107 parasites per mL (0.125 µL – 6.25 x 103 parasites, 0.25 µL – 1.25 x 104 parasites, 0.5 µL – 2.5 x 104 parasites, 0.75 µL – 3.75 x 104 parasites, 1 µL – 5 x 104 parasites, 5 µL – 2.5 x 105 parasites, and 10 µL – 10 x 105 parasites) (Figure 2), and its sensitivity to detect concentrations ≥103 was demonstrated which is above the number of parasites (L. infantum) per mL in natural sand flies (phlebotomines, Lu. longipalpis) after the infection where it was identified +105 parasites per mL [10].

For the data in Figure A,B, and C:  please indicate the size (along the y axis) of the markers (using bp); in the legend, it indicates that 100 bp marker was used…does this indicate that the visible spots in the electrophoresis are 10-20 bp?  Please indicate that Figure 2 are results of electrophoresis.  Indicate the size of the spots to the right of the gel.

Reply: We included the information in Figure 2.

I am very unclear how the percentages were calculated (line 146 and in Table 1). This should be clarified.  

Reply: Dear reviewer, we added the following excerpt in the Table 1 legend:

*, the percentage of females and males was calculated by the following formula = (number of females or males)/(total number of Lutzomyia longipalpis collected in each municipality). **, the infection minimum rate was calculated using the following formula: the number of positive groups (pools) x 100/Total phlebotomines processed [11,12].

Line 208: add ‘thus’ following the word environments to make this sentence clearer.

Reply: We included the word as suggested by the reviewer.

Line 213:  the data are not sorted by year so not sure how the authors can indicate ‘perennial’.

Reply: Dear reviewer, we thank you for your essential contribution to our study. Really, in our study protocol is not evident the evolution of the cases in Brazil using our data collection. However, we had this information available from the Brazilian Health Ministry. In this context, we included the following statement in the abstract and conclusions:

Abstract:

The L. infantum infection in the phlebotomines in the regions investigated seems to be active and perennial, with an intense transmission reported by the Brazilian health authorities.

Conclusions:

Phlebotomine infection by the Leishmania genus in the regions investigated in the state of São Paulo was seen to be active and perennial, with an intense transmission reported by the Brazilian health authorities. The method used to extract DNA from the pools of 10 phlebotomines (Cation paper) and the PCR reactions were seen to be useful due to their sensitivity, specificity, practicality, and faster analysis, when compared to the individual analysis method. This resource can be used on a large scale in the epidemiological surveillance of Leishmaniasis, enabling a higher number of analyses and the optimization of human resources.

Line214:  replace ‘was’ with ‘were’

Reply: We corrected the word as suggested by the reviewer.

Line 215: the authors give no information about cost in this manuscript so not sure they can indicate ‘lower cost’.  A cost analysis is thus needed to make this statement.

Reply: Dear reviewer, we performed some corrections in the text as we decided to include only a minor modification in the excerpt described by you as follows:

The method used to extract DNA from the pools of 10 phlebotomines (Cation paper) and the PCR reactions were seen to be useful due to their sensitivity, specificity, practicality, and faster analysis, when compared to the individual analysis method. This resource can be used on a large scale in the epidemiological surveillance of Leishmaniasis, enabling a higher number of analyses and the optimization of human resources.

The PCR showed great sensitivity and specificity due to the remarkable capacity to identify a low amount of DNA from the Leishmaniaachieved after the extraction from the L. longipalpis sand flies. Also, the study protocol is easily handled when compared with the individual analysis method (desiccation of the insect digestive system and microscopic examination), which is time-demanding.

In addition, we removed the citation of cost because both protocols (DNA extraction/PCR and individual analysis) had different prices, and it was not evaluated in the present study.

In addition, for the title, change ‘in’ to ‘using’

Reply: We corrected the word as suggested by the reviewer.

The genus Leishmania should be in italics every where in the manuscript.

Reply: We used the Italic type as recommended.

Should ‘Leishmaniasis’ be capitalized or lower case throughout the manuscript?  ….should be consistent.

Reply: We capitalized the word in the text.

Author Response

Comments and Suggestions for Authors 3

The authors present a new method for the detection of L. infantum DNA in pools of in Lu. Longipalpis sand flies in two areas in the state of Sao Paulo differing in their parasite transmission cycle. I think they developed a nice test for studying infection rates in sand flies which can be applied (and perhaps adapted to other vectors or Leishmania parasites) in epidemiological studies in leishmaniases. The manuscript needs however some improvement.

Reply: The authors thank the reviewer for contributing to our study. We tried our best to reply to all the comments and perform all the text corrections.

First of all, the English needs to be improved by a native speaker.

Reply: The English writing was edited. In addition, after acceptance, minor corrections are made by the editorial staff to avoid minor grammatical errors.

Title:

Since the authors have only tested for L. infantum, this should be stated in the title. I would suggest:

“Monitoring Leishmania infantum infections in female Lutzomyia longipalpis by using DNA extraction on Cation paper and PCR pool testing”

Reply: The title was corrected: “Monitoring Leishmania infantum infections in female Lutzomyia longipalpis by using DNA extraction on Cation ex-change paper and PCR pool testing.”

Abstract:

Delete the sentence in lanes 25 to 28, which describes the methodology in detail not needed in the abstract. Also delete the sentence in lanes 32 to 34 which should be rather in the introduction explaining why these two areas were chosen for this study.

Reply: The authors thank the reviewer, and we excluded the following excerpt:

“The Cation exchange paper containing the phlebotomines was kept at 37 ºC for 15 min, and the excess carcass was removed. Next, the material was homogenized in a vortex using 500 µL ultrapure water for 20 seg and centrifuged at 10,000 r.p.m. for 15 min. The sediment was then subjected to DNA extraction.”

In addition, the information in the following excerpt, “Samples with identification for L. infantum were found in the two cities with the intense transmission of this disease with a 0.22% infection minimum rate. The L. infantum infection in the phlebotomines in the regions investigated seems to be active and perennial, with an intense transmission reported by the Brazilian health authorities.” was included in the other topics of the study.

Introduction:

I think the introduction is too short. The information about the study area given in two paragraphs in the Discussion (lanes 217 to 225) should be moved to the introduction. Also, I would like to get a definition about what the difference between “intense” and “dog” transmission is. The choice of the two study areas needs to be explained.

Reply: We corrected the introduction. In addition, we include more information about the suggested terms in the text.

M&M:

The methods were clearly described. But why the authors did put the word “animals” into Figure 1?

It should be replaced by “sand fly specimen” or “individuals”.

Reply: We corrected figure 1.

Results:

This section should start with the information given in paragraphs 2 and 3 (lanes 151 to 159). Only than the paragraph about the two different paper filters should follow.

Reply: We included the correction.

If I understood correctly, there were two PCRs used, one general for detecting Leishmania spp. And a second for identification of L. infantum. Where there any samples positive for the Leishmania spp. but not for L. infantum? Is Lu. Longipalpis a specific vector for L. infantum or can these sand flies also harbour other Leishmania species?

Reply: Dear reviewer, the L. infantum was tested at the specie level because it is responsible for causing Visceral Leishmaniasis in Brazil. However, we also tested the genus to demonstrate that the technique can be applied in other places where different species are liable to cause the disease.

Discussion:

I think the new method is sufficiently discussed.

But I have a problem with the sentence in lanes 226 to 228 stating that “this study revealed a coherence between test positivity and the illness epidemiology…”. First, just two of the pools were positive, and secondly, the number sand flies differed considerably between the two study areas. Whereas 920 females (92 pools) were collected in the area with “intense” transmission, only 240 females (14 pools) were collected in the area with “dog” transmission. I think investigation of more sand flies especially from the second area is needed to conclude on whether infection rates in these regions are related with “illness” epidemiology. So far, it can be only a hypothesis that can be discussed.

Reply: We included the following excerpt in the text:

“Our results revealed positive test results for municipalities with perennial human transmission and negative where only canine transmission was found. However, we had a low number of pools in the cities where only canine transmission was found; and only two pools were positive where perennial human transmission occurred. In such context, our results demonstrated that DNA extraction and PCR were efficacy in evaluating the presence of the parasite; however, we cannot prove the perennial human transmission and canine transmission status in the cities where the sand fly specimen were collected due to several factors, such as the low number of samples collected.”

Round 2

Reviewer 1 Report

Basically, the manuscript was not improved. 

Author Response

Revision

Comments and Suggestions for Authors 1

Basically, the manuscript was not improved.

Reply: The authors thank the reviewer for correcting the first version of the manuscript. We tried our best to improve it according to the reviewers’ comments. Also, we are submitting a new version of the manuscript based on the remarks of reviewers 1 and 2.

Reviewer 2 Report

I wish to thank the authors for their extensive revisions and clarifications of this interesting manuscript.  I, however, strongly recommend that they change all "cation paper' to 'cation exchange paper' throughout the manuscript and in the title; this will be less confusing for the reader.  Also indicate anion exchange paper rather than anion paper.  Also, please change the word 'in' to 'using' in the title as previously requested.  For the legend in Figure 2, please clarify what is in the 'M' lane (legend indicates 100 bp molecular weight markers yet both 120 bp and 145 bp are indicated to the left of the bands).  

Author Response

Comments and Suggestions for Authors 2

I wish to thank the authors for their extensive revisions and clarifications of this interesting manuscript. I, however, strongly recommend that they change all "cation paper' to 'cation exchange paper' throughout the manuscript and in the title; this will be less confusing for the reader.

Reply: Dear reviewer, we included the correct information.

Also indicate anion exchange paper rather than anion paper.

Reply: Dear reviewer, we included the correct information.

Also, please change the word 'in' to 'using' in the title as previously requested.

Reply: Dear reviewer, thank you so much. Our apologies for the error.

In addition, the other reviewer suggested a new title: “Monitoring Leishmania infantum infections in female Lutzomyia longipalpis by using DNA extraction on Cation ex-change paper and PCR pool testing.”

For the legend in Figure 2, please clarify what is in the 'M' lane (legend indicates 100 bp molecular weight markers yet both 120 bp and 145 bp are indicated to the left of the bands).

Reply: We included the following information:

“M, 100 bp molecular weight marker (100 bp DNA Ladder) and the DNA fragments obtained after the DNA amplification for Leishmania spp. Identification (120 bp) and L. infantum specie (145 bp)”.